# Effects of Replacing Yellow Corn with Olive Cake Meal on Growth Performance, Plasma Lipid Profile, and Muscle Fatty Acid Content in Broilers

**DOI:** 10.3390/ani11082240

**Published:** 2021-07-29

**Authors:** Ahmed Saleh, Mohammed Alzawqari

**Affiliations:** 1Department of Poultry Production, Faculty of Agriculture, Kafrelsheikh University, Kafrelsheikh 333516, Egypt; m.alzawqari@gmail.com; 2Department of Animal Production, Faculty of Agriculture and Food Sciences, Ibb University, Ibb 70270, Yemen

**Keywords:** olive cake meal, growth performance, plasma lipid, muscle fatty acids content, broilers

## Abstract

**Simple Summary:**

Yellow corn is a grain frequently utilized in broiler diets. Moreover, the new era of the corn-ethanol industry for biofuel production has increased the divergence of energy applications of corn. These uses have considerably increased corn prices over the past few years. Using byproducts may be one of solutions to this problem. Olive cake meal (OCM) is a byproduct obtained from olive oil factories after olive oil is extracted. It has a high nutritional value, especially in regard to metabolizable energy. OCM has been used successfully in livestock and poultry feeding as an alternative to energy sources such as corn, without any adverse effects on performance. Hence, the current study investigated the effects of replacing yellow corn with OCM on the growth, nutrient utilization, selected blood parameters, and muscle fatty acid profile of broilers. Four hundred and eighty one-day-old male broiler chickens (Ross 308) were divided into four experimental groups (a control group, and groups with 5%, 10%, and 20% of corn replaced with OCM). The results revealed that replacing 10% of corn with OCM in the diet improved growth performance and reduced abdominal fat and liver malondialdehyde (MDA) and increased high density lipoprotein (HDL).

**Abstract:**

The current study focused exclusively on evaluating the effects of replacing corn with olive cake meal (OCM) in the diet of broilers on their growth performance, abdominal fat, selected plasma parameters, and muscle fatty acid (FA) content. A total of 480 one-day-old male broiler chickens (Ross 308) were divided into four treatment groups with 12 replicates/treatment. The control group was fed the base diet, whereas the second to fourth groups were fed diets of corn with 5%, 10%, and 20% contents of OCM, respectively. Broilers fed with the 5% and 10% OCM diets showed better body weight (*p* = 0.04) and feed conversion ratio than the 20% OCM group (*p* < 0.048). Both nitrogen retention and ether extract digestibility were not improved by replaced corn with OCM. Replacing corn with OCM led to a decreased abdominal fat percentage (*p* = 0.023) compared with the control group. Birds in the OCM groups showed the lowest total cholesterol values (*p* = 0.038). The breast muscle (musculus pectoralis superficialis) content of oleic and linoleic, linolenic, and arachidonic acids was significantly high in birds fed with OCM diets. However, their palmitic acid level was significantly decreased. Vitamin E was increased by increasing the OCM level. Thus, we concluded that replacing corn with OCM, especially at a 10% level, is more effective than other replacement levels in improving growth performance, plasma lipid profile, and muscle FA content, as well as in causing a reduction in abdominal fat in broilers.

## 1. Introduction

Recently, the increased use of yellow corn for ethanol production has consumed a significant amount of corn, which has had a large effect on the price of corn. This has had a major impact on poultry production as well. For example, in broiler production, yellow corn constitutes ~60–70% of the poultry diet. A decrease in the availability of corn and an increase in the price of diets have a major effect on broiler production. Therefore, we must find new alternative ingredients that can be used as a substitute for yellow corn partially or totally, and these alternatives must not have a negative effect on the growth performance of broilers [1,2]. It is worth noting that significant attempts have been made to find alternative and long-term protein sources for broiler diets [3]. Given the above, olive cake meal (OCM) is a byproduct of olive oil extraction that is used in poultry feed. However, during the oil extracting methods, we cannot extract all of the oil from the seeds and, thus, this byproduct was higher in terms of its fat content. A few compounds that may have antioxidant properties are included in this byproduct [4]. OCM is high in oil and can be used as a source of energy and antioxidant compounds. The levels of any new ingredient in a diet are another essential factor, which affects the feed consumption and feed conversion ratio. It also helps to reduce ration costs, as well as the feeding costs for animals such as sheep, poultry, and aquatic organisms such as *Cyprinus carpio*. The variability of its chemical composition is one of the most critical limitations of OCM [5].

OCM is also high in lignins. It contains xyloglucan (a form of non-starch polysaccharide), which has been linked to anti-nutritional effects in monogastric organs such as those found in poultry, limiting its use in poultry feed [6]. Moreover, the presence of significant amounts of anti-nutritional phenolic compounds such as tannins in the diet, when combined with protein and carbohydrates, may limit nutrient availability by reducing the performance of digestive enzymes [7,8]. Furthermore, one of the drawbacks of OCMs is their high moisture and fat content, making consumption and storage difficult [9]. However, some processing methods used in olive oil extraction (e.g., emerging technologies including microwave (MW), pulsed electric field (PEF), and ultrasound (US)) may improve the digestibility of OCM and bring about a reduction in these anti-nutritional factors [2].

As well as having a high content of non-starch polysaccharides (NSPs), OCM has a high nutritional value (crude fat, 13–18%; crude proteins, 9–10%) [10,11,12]. Additionally, vitamin E, calcium, iron, potassium, magnesium, sodium, and phosphorus are all contained within OCM [13]. Large amounts of OCM are generally produced during the extraction of oils from olives. In addition, Al-Harthi et al. [14] showed that OCM can be found worldwide and can be used as a plant source for broiler feeds at a fair price in several countries. Moreover, Sadeghi et al. reported that replacing corn with OCM is an effective way to recycle waste materials in sheep diets [15]. There is a considerable amount of residual oil (6.8%) in OCM, which can be used as a source of additional energy.

Furthermore, broiler tissue fatty acid (FA) profiles are affected by the oleic, linoleic, and linolenic acids in OCM [2]. In this respect, Al-Harthi and Sateri recently suggested the use of up to 10% of OCM in the total formula in broiler diets [11,16]. Furthermore, broiler performance has no adverse effect when the feed is replaced with up to 150 g OCM/kg [17,18]. OCM has been recommended for broiler chicken nutrition at a rate of 5–10%. Researchers found that feeding with 10% of OCM from olive oil extracts had no negative impact on the weights of internal organs, the gastrointestinal tract (GIT), carcass parts, or the dressing percentage. Nevertheless, birds fed with a 10% OCM diet showed the best live body weight (BW) [12,17,19]. Sayehban et al. also reported on the inclusion of olive pulp byproducts in the diets of broilers to increase the carcass and other organ weights [20]. Papadomichelakis et al. recently demonstrated that broilers fed with 50–80 g/kg of dried olive pulp showed better growth performance (body weight and feed conversion ratio) and meat quality (reduced oxidative stability and better meat color) [21]. According to reports by Saleh et al. and Cayan and Erener [2,22], adding OCM byproducts to poultry diets improved total serum protein, albumin, HDL cholesterol concentration, and resulted in an increased HDL/LDL ratio, correlated with a lower total cholesterol and LDL content. Furthermore, feeding broilers a diet containing OCM increased the polyunsaturated fatty acid (PUFA) content in the breast muscle [23].

Further research into the growth performance and other physiological traits of meat quality and fatty acids contents of commercial broilers fed with olive byproduct meal diets appears to be needed in light of these observations. The purpose of the present study was to evaluate the effects of the replacement of corn with olive cake meal (OCM) in broiler diets in terms of growth performance, abdominal fat, selected plasma parameters, and muscle FA content.

## 2. Materials and Methods

The study was approved by the Ethics Committee of the Local Experimental Animals Care Committee and was conducted following the guidelines of Kafrelsheikh University, Egypt (Number 4/2016 EC). All precautions were followed to decrease suffering during the entire experimental period.

### 2.1. Experimental Design

A total of 480 one-day-old male broiler chickens (Ross 308) were placed inside a room equipped with 48 pens (10 birds each) (4 treatments/12 replicates each; stocking density was ten birds/m^2^) with a chain feeder system and an automatic nipple cup drinker. The sequencing of the pens followed the first replicates from each treatment, then the second replicates for each treatment, until reaching 12 replicates from each treatment. The experimental treatments consisted of diets formulated based on the nutritional requirements of the NRC [24] for male broilers, with a three-phase feeding system (starter diets from 0–10 days as crumble form, grower diets from 11–24 day as pellet form, and finisher diets from 25–35 day as pellet form, respectively) (Table 1), with all of the diets provided in pelted form. The first group of birds served as the control group and was fed with a base diet. The second to fourth groups were fed diets that replaced corn with OCM at 5%, 10%, and 20%. The diets and OCM were provided by Al-Sabeel Al-Gadidah Company, Tanta, Al-Gharbia, Egypt.

The chemical composition values used for OCM were analyzed in the laboratory of feed analysis at Kafrelsheikh University, Egypt and the values were recorded by national research council (NRC) [24] and shown in Table 2. The metabolizable energy content of OCM was calculated with the following equation:Men = 26.7 × DM + 77 × EE − 51.22 × CF
where: DM: dry matter, %. EE: ether extract, %. CF: crude fiber, %.

The diets were provided to the birds ad libitum. The feed trial took place in a temperature-controlled room with an 18-h light and 6-h dark cycle, at a temperature of starting from 32 ± 1 °C and decreasing by 1 °C every three days until reaching 24 ± 1 °C, which was maintained until the end of the experiment (35 days), and proportional humidity was kept between 50% and 70%. Throughout the experimental phase, the health status and mortalities were measured regularly each day.

### 2.2. Growth Performance and Carcass Parts

Growth performance (body weight and feed intake and feed conversion ratio were measured every week on a group as 12 pens) throughout the experimental period. At 35 days, all birds were weighed individually and sorted from smallest to heaviest. Then, 48 birds (1 bird per replicate; 12 birds per treatment) were slaughtered and then dissected to measure the weights of the breast muscle, the thigh muscle, the liver, and the abdominal fat according to Saleh et al. [25,26].

### 2.3. Nitrogen Retention and Ether Extract Digestibility

In the last three days of the experiment, excreta were gathered and weighted from 12 birds per treatment, where broilers were housed individually in special metabolic cages (40 × 40 × 50 cm) for digestibility tests. During these three days, the birds and feed intake were weighted daily, and extracted faces were collected, weighted, and stored in a freezer. After the digestibility experiment period, all samples were dried in a drying oven at 60 °C for 24 h. The whole dried samples were then homogenized. Samples were taken and finely ground for analysis according to the Association of Official Analytical Chemists (AOAC). The crude protein concentration in the diet and excreta was gauged to determine nitrogen retention using the Kjeldahl method, and ether extract digestibility was gauged by the Soxhlet method (AOAC 945.38 F and 920.39 C, respectively). The calculation was as follows: Nitrogen retention (%) = (total nitrogen intake − total nitrogen excreted)/total nitrogen intake × 100.

### 2.4. Selected Plasma Parameters

At 35 days, blood samples from 48 birds (1 bird per replicate; 12 birds per treatment) were collected from the wing vein immediately before slaughtering, gathered into heparinized test tubes, and then rapidly centrifuged (3000 rpm for 20 min at five °C) to separate the plasma. Plasma was stored at −20 °C pending analysis. Plasma total cholesterol, high-density lipoprotein (HDL), glutamic oxaloacetic transaminase (GOT), total protein, albumin and globulin, and uric acid were spectrophotometrically assessed (Spectronic 1201; Milton Roy, Ivyland, PA, USA) using commercial kits (Cell Biolabs Inc., San Diego, CA, USA) according to the manufacturer’s instructions.

### 2.5. Breast Meat Fatty Acid Profile

The analysis of breast muscle (musculus pectoralis superficialis) FAs was conducted on 48 birds (1 bird per replicate; 12 birds per treatment) from the breast muscle (superficial pectoral muscle) using gas-liquid chromatography (GLC) (Model GC-14A, Shimadzu Corporation, Kyoto, Japan) with a flame ionization detector and apolar capillary column (BPX70, 0.25; SGE Incorporated, Washington, DC, USA) according to the procedure of Saleh et al. [27]. Lipid peroxidation was evaluated by assaying the level of malondialdehyde (MDA) in the liver by using kits from Cell Biolabs Inc. (San Diego, CA, USA) and muscle vitamin E according to Ohkawa et al. [28].

### 2.6. Statistical Analysis

The differences between the treatment and control groups were analyzed with a General Liner model using SPSS (Version 17.0, Chicago, IL, USA). One-way ANOVA was applied to determine the effects of replacing corn with OCM, in which pens were the statistical units for performance parameters, birds for the carcass, organ weights, and samples for biochemical and other parameters. Duncan’s new multiple range test was used to identify which treatment conditions were significantly different from each other at a significance level of *p* < 0.05.

## 3. Results

### 3.1. Bird Performance and Organ Weights

The effect of replacing corn with OCM in broilers’ diets on performance, including final body weight (BW), feed intake (FI), and feed conversion ratio (FCR) during the experiment, is shown in Table 3. Broilers fed with dietary corn replaced with 5% or 10% OCM had better body weights than the 20% OCM-replaced group (*p* < 0.040). However, the lowest feed intake by replaced 20% of OCM. Additionally, there were no differences among the groups showed in feed conversion ratios compared with control group. Likewise, there were no differences among the groups (*p* = 0.47), concerning the carcass and organ weights of broiler chickens at 35 d of age. The use of OCM as a replacement leads to a deceased abdominal fat percentage (*p* = 0.023) compared with the control group. Both nitrogen retention and ether extract digestibility were not influenced by replaced corn with OCM.

### 3.2. Selected Plasma Parameters

The results concerning the effect of replacing corn with OCM in broiler diets on selected plasma parameters during the experimental period are shown in Table 4. The contents of GOT, uric acid, total protein, and albumin in the plasma were not affected by the content of OCM in the diet. An exception was the total cholesterol level in the plasma. This decreased significantly (*p* = 0.038) when replacing corn with 5%, 10%, and 20% OCM in the broilers’ diets. In addition, the highest HDL value (*p* = 0.022) was found in the 20% OCM group.

### 3.3. Fatty Acid, Vitamin E, and MDA Content

The effects of replacing corn with OCM in broiler diets on breast muscle FA content are presented in Table 5. The levels of oleic and linoleic, linolenic, and arachidonic acids in the muscles increased significantly in broilers in the OCM groups. However, the palmitic acid level was significantly decreased, whereas the myristic, palmitoleic, stearic, vaccenic, eicosapentaenoic, docosapentaenoic, and docosahexaenoic acids were not altered by the dietary replacement of corn with OCM. Muscle vitamin E and liver MDA were significantly influenced by the dietary replacement of corn with OCM. The vitamin E value was increased by increasing the OCM replacement level (*p* = 0.021). The liver MDA values were decreased by increasing the OCM replacement level (*p* = 0.034). The control diet was showed the highest value in terms of MDA.

## 4. Discussion

Today, the use of low-cost resources in the poultry diet is one way to reduce production costs, especially in intensive rearing systems. As a result, the use of agricultural and conversion industry byproducts, especially oilseeds, has a unique position in terms of providing energy to humans, livestock, poultry, and aquatic animals [1,2]. The utilization of olive byproducts (for example, olive cake meal) in broiler diets represents a good way of recycling these waste products. However, there is a need to formulate optimized ratios for different uses to avoid metabolic disorders caused by unbalanced ratios of energy and protein and to reduce the taste factors which might limit feed intake and thus broiler growth performance, leading to low profitability. Most of the studies that have been conducted in this area have focused on the feeding of OMC to broilers and its effect on growth performance and meat quality. In the current study, we aimed to find the best replacement levels of yellow corn using OCM, which would not lead to any negative effects on broiler production.

The present study showed that replacing 10% of a corn broiler diet with OCM during the experimental period did not affect FI and FCR compared to the control group. This is in agreement with the results of several studies [17,29,30]. Moreover, a 20% OCM content in the broiler corn diet resulted in a reduction in BWG but a 10% OCM content. By contrast, Papadomichelakis et al. found that even a low dietary dried olive pulp content of 50 to 75 g/kg could reduce the BW gain and feed efficiency early in broilers [21]. Thus, the high fiber content and anti-nutritional factors present when high levels of OCM were used can be blamed for the reduction in broiler performance and did not improved nitrogen retention and ether extract digestibility [14,30]. The carcass and organ weights in this study were similar between treatment groups.

Similar to our findings, de Oliveira et al. reported that no significant effects on the carcass and organ weights were observed when including OCM in the diets of broilers at 35 days of age [31]. In previous OCM studies on broiler diets using contents of 5% to 10% with and without enzyme additives, carcass and organ weights were not affected [2,14]. OCM also has no adverse effects on carcass traits and internal organs when used up to 10% [11].

Replacing corn with OCM in the current study reduced the abdominal fat. Similarly, Al-Harthi et al. and Saleh et al. reported that using OCM in the broiler diet reduced abdominal fat [2,11]. Broiler feed was shown to increase the body fat content, which could explain why the control birds had higher abdominal fat levels than broilers fed with the OCM diet. Previous research has shown that broilers fed with unsaturated FA-rich diets have a lower abdominal fat content [32,33,34] and total carcass fat [35] deposition than broilers fed with saturated FA-rich diets. Including OCM in diets was shown to increase the unsaturated to saturated FA ratios in all birds studied [17,36].

The plasma concentrations of GOT, uric acid, total protein, and albumin were not affected by the OCM-replaced diet. Total plasma cholesterol levels were found to decrease significantly (*p* = 0.038) when corn was replaced with OCM in the diets of broilers. Furthermore, broilers fed with the OCM-replaced diet had the highest HDL values (*p* = 0.022). Similarly, Sateri et al. found that supplementing birds with OCM did not affect total serum protein and albumin levels [16]. Because of OCM’s unsaturated and polyunsaturated FAs, most studies have shown that replacing corn with OCM in broiler diets reduces the total lipid content and increases the HDL:LDL ratio [16,29,37], with researchers finding that adding OCM at 10% in broiler diets leads to a substantial decrease in blood cholesterol and triglyceride levels. Saleh et al. observed that feeding with OCM increased plasma HDL levels but decreased plasma cholesterol levels [2]. This may be due to the benefits of high level of unsaturated fatty acids.

Regarding the effects of corn replacement with OCM in broiler diets on muscle FA content, in the present study muscle unsaturated FAs were increased by the presence of OCM. The high content of unsaturated and polyunsaturated FAs in OCM can lead to the effects observed in OCM diets [17,31]. In addition, Chamruspollert and Sell reported that the presence of olive oil in OCM can cause modifications in unsaturated FAs by inhibiting the delta-9 desaturase enzyme system, which is responsible for desaturating saturated FAs and converting them to unsaturated FAs in the muscles [38].

In the current study, feeding with OCM increased the vitamin E level of the muscles in birds, which may lead to a reduction in the lipid peroxidation process in the birds and, as a result, reduce the oxidation status [36]. Since the essence of the diet has a direct relationship with oxidative damage in cultured organisms, oxidative focus occurs when the development and removal processes of free radicals (ROS) are unbalanced [39,40]. Reactive oxygen species (ROS) scavengers, such as superoxide dismutase, glutathione peroxidase, and catalase, protect body tissues from oxidative stress [41]. Malondialdehyde is a by-product of lipid peroxides, which have a high reactive oxygen content, and can also damage DNA cell and cytoplasm proteins [42,43]. OCM feed led to decreased levels of malondialdehyde, as reported by Saleh et al. [2], and previous studies have reported an increased antioxidant reaction to OCM [44,45,46].

## 5. Conclusions

In conclusion, the replacement of 10% of corn with olive cake meal (OCM) is suitable for broiler diets, resulting in improved growth performance, reduced abdominal fat, decreased plasma cholesterol, and increased oleic acid as monounsaturated fatty acids and linolenic acid as polyunsaturated fatty acid contents.

## Figures and Tables

**Table 1 animals-11-02240-t001:** Composition of the experimental starter (1–11 day), grower (11–24 day), and finisher (25–35 day) diets *.

Ingredient, g/kg	Control			Replaced 5%			Replaced 10%			Replaced 20%		
	Starter	Grower	Finisher	Starter	Grower	Finisher	Starter	Grower	Finisher	Starter	Grower	Finisher
Yellow corn	530	580	640	503.5	551	608	472	522	576	424	464	512
Soybean meal, 44%	360	304	227	359	304	226	361	303	227	360	303	226
Corn gluten meal, 62%	46	48	61	47	47	62	48	48	61	46	48	62
Soybean oil	24	29	32	24	30	32	26	30	32	24	30	32
Dicalcium phosphate	16	15	15	16	15	15	16	15	15	16	15	15
Dl Methionine, 99%	2	1.8	1.2	2	1.8	1.2	2	1.8	1.2	2	1.8	1.2
l-Lysine HCl, 98%	1.3	1.4	2.4	1.3	1.4	2.4	1.3	1.4	2.4	1.3	1.4	2.4
l-Threonine	0.5	0.3	0.1	0.5	0.3	0.1	0.5	0.3	0.1	0.5	0.3	0.1
CaCo_3_	12	12	11	12	12	11	12	12	11	12	12	11
NaCl	3.5	3.5	3.5	3.5	3.5	3.5	3.5	3.5	3.5	3.5	3.5	3.5
Premix **	3	3	3	3	3	3	3	3	3	3	3	3
NaCo_3_	1.5	1.5	1.6	1.5	1.5	1.6	1.5	1.5	1.6	1.5	1.5	1.6
K_2_Co_3_	0.2	0.5	2.2	0.2	0.5	2.2	0.2	0.5	2.2	0.2	0.5	2.2
Olive cake meal	0	0	0	26.5	29	32	53	58	64	106	116	128
	Chemical Analysis											
Crude protein, %	23.00	21.00	19.00	23.05	21.03	19.02	23.04	21.04	19.02	23.05	21.074	19.03
AME kcal/kg	2950	3045	3150	2950	3041	3142	2950	3040	3141	2950	3040	3141
Ca, %	0.95	0.896	0.864	0.95	0.895	0.864	0.95	0.897	0.864	0.96	0.894	0.864
Available P, %	0.422	0.408	0.388	0.422	0.408	0.388	0.422	0.408	0.388	0.422	0.408	0.388
Crude fiber, %	3.444	3.547	3.333	3.786	3.992	3.909	4.311	4.3272	4.334	5.444	5.131	5.509
Na, %	0.193	0.193	0.196	0.193	0.193	0.196	0.193	0.193	0.196	0.193	0.193	0.196
Cl, %	0.250	0.250	0.25	0.250	0.250	0.249	0.250	0.2542	0.249	0.250	0.248	0.2497

* The basal diet fed to the chicks was formulated to meet the NRC recommendations for broiler chickens; ** Premix (Hero mix^®^, Hero pharm, Cairo, Egypt). Composition (per 3 kg): Vitamin A, 12,000,000 IU; vitamin D3, 2,500,000 IU; vitamin E, 10,000 mg; vitamin K3, 2000 mg; vitamin B1, 1000 mg; vitamin B2, 5000 mg; vitamin B6, 1500 mg; vitamin B12, 10 mg; niacin, 30,000 mg; biotin, 50 mg; folic acid, 1000 mg; pantothenic acid, 10,000 mg; manganese, 60,000 mg; zinc, 50,000 mg; iron, 30,000 mg; copper, 4000 mg; iodine, 300 mg; selenium, 100 mg; and cobalt, 100 mg. There were no feed additives added to the diets.

**Table 2 animals-11-02240-t002:** Nutrient composition of olive cake meal (% DM).

Nutrients	Olive Cake Meal
Crude protein, g/kg	8.2
Metabolizable energy, MJ/kg	13.933
Calcium, g/kg	0.021
Total phosphorus, g/kg	0.46
Ether extract, g/kg	17.8
Crude fiber, g/kg	11.55
Fatty acids, g/100 g fatty acids	
Myristic acid	0.98
Palmatic acid	7.3
Palmitoleic acid	0.86
Stearic acid	4.4
Oleic acid	72.1
Linoleic acid	9.5
Linolenic acid	4.5
Saturated fatty acids, %	13.54
Unsaturated fatty acids, %	86.46
Polyunsaturated fatty acids, %	14.0

**Table 3 animals-11-02240-t003:** The effects of replacing corn with olive cake meal (OCM) on broilers’ growth performance and organ weights.

Item	Experimental Diets				SEM	*p*-Value
	Control	Replaced 5%	Replaced 10%	Replaced 20%		
Initial body weight, g	42.2	42.0	42.2	42.3	0.3	0.881
Body weight 35 d, g	2089 ^a,b^	2186 ^a^	2164 ^a^	2064 ^b^	26	0.040
Feed intake 35 d, g	3384 ^a^	3355 ^a,b^	3335 ^a,b^	3314 ^b^	54	0.035
FCR, 35 d	1.62 ^a^	1.54 ^b^	1.54 ^b^	1.61 ^a,b^	0.02	0.048
Nitrogen retention, %	71	73	73	72	5	0.097
Ether extract digestibility, %	41	43	43	44	3	0.124
	Organ weights, % body weight					
Carcass	66.5	66.2	67.4	67.7	2.5	0.470
Breast muscle	22.42	22.51	22.53	22.32	2.70	0.743
Thigh muscle	16.46	16.38	16.27	16.41	0.98	0.349
Liver	2.21	2.22	2.16	2.23	0.11	0.528
Abdominal fat	2.17 ^a^	1.90 ^b^	1.87 ^b^	1.72 ^b^	0.18	0.023

^a,b^ Means within the same row with different superscripts differ (*p* < 0.05). Results are presented as means ± SEM. Pens were used as sample units for growth performance (initial body weight, body weight, feed intake and FCR), whereas birds were used as samples units for organ weights and digestibilities (12 bird/treatment). SEM; standard error of mean.

**Table 4 animals-11-02240-t004:** The effects of replacing corn with olive cake meal (OCM) on plasma parameters in broilers.

Item	Experimental Diets				SEM	*p*-Value
	Control	Replaced 5%	Replaced 10%	Replaced 20%		
GOT, mg/dL	243	231	234	225	32	0.632
Uric acid, mg/dL	6.13	6.22	6.62	6.42	0.34	0.514
Total protein, mg/dL	3.33	3.52	3.55	3.48	0.41	0.711
Albumin, mg/dL	1.95	2.11	2.11	2.17	0.08	0.232
Total cholesterol, mg/dL	151 ^a^	133 ^b^	131 ^b^	129 ^b^	14	0.038
HDL-cholesterol, mg/dL	78 ^c^	82 ^b^	86 ^b^	91 ^a^	12	0.022

^a,b,c^ Means within the same row with different superscripts differ (*p* < 0.05). Results are presented as means ± SEM. 12 samples were used per treatment. SEM; standard error of mean.

**Table 5 animals-11-02240-t005:** The effects of replacing corn with olive cake meal (OCM) on fatty acid profiles in broilers.

Item	Experimental Diets				SEM	*p*-Value
	Control	Replaced 5%	Replaced 10%	Replaced 20%		
Myristic acid (C14:0), mg/100 g fat	1.42	1.38	1.38	1.37	0.02	0.571
Palmitic acid (C16:0), mg/100 g fat	23.16 ^a^	22.04 ^a,b^	21.16 ^b^	21.03 ^b^	3.78	0.041
Palmitoleic acid (C16:1), mg/100 g fat	5.52	5.51	5.55	5.48	0.32	0.753
Stearic acid (C18:0), mg/100 g fat	8.71	8.33	8.46	8.31	0.88	0.431
Oleic acid (C18:1 n − 9c), mg/100 g fat	40.05 ^b^	42.59 ^a,b^	44.78 ^a^	45.15 ^a^	14.00	0.039
Vaccenic acid (C18:1 n − 7), mg/100 g fat	5.51	5.72	5.47	5.66	0.54	0.632
Linoleic acid (C18:2 n − 6), mg/100 g fat	8.31 ^b^	9.15 ^a,b^	9.98 ^a^	10.10 ^a^	1.24	0.043
Linolenic acid (ALA, C18:3 n − 3), mg/100 g fat	0.76 ^b^	0.79 ^a,b^	0.82 ^a^	0.86 ^a^	0.06	0.047
Arachidonic acid (AA, C20:4 n − 6), mg/100 g fat	2.13 ^b^	2.76 ^a,b^	2.87 ^a^	2.89 ^a^	0.08	0.043
Eicosapentaenoic acid (EPA, C20:5 n − 3), mg/100 g fat	0.033	0.037	0.036	0.036	0.002	0.821
Docosapentaenoic acid (DPA, C22:5 n − 3), mg/100 g fat	0.224	0.214	0.211	0.217	0.005	0.712
Docosahexaenoic acid (DHA, C22:6 n − 3), mg/100 g fat	0.971	0.983	0.979	0.993	0.009	0.069
Muscle vitamin E, gm/100 g muscle	0.26 ^c^	0.29 ^b^	0.32 ^a^	0.40 ^a^	0.018	0.021
Liver MDA, nmol/g	22 ^a^	19 ^b^	15 ^c^	14 ^c^	1.92	0.034

^a,b,c^ Means within the same row with different superscripts differ (*p* < 0.05). Results are presented as means ± SEM. 12 samples were used per treatment. SEM; standard error of mean.

## Data Availability

All data sets collected and analyzed during the current study are available from the corresponding author on reasonable request.

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
