# Peer review of "Effects of Replacing Yellow Corn with Olive Cake Meal on Growth Performance, Plasma Lipid Profile, and Muscle Fatty Acid Content in Broilers"

_animals, 2021, doi:10.3390/ani11082240_

Round 1

Reviewer 1 Report

The manuscript by Ahmed et al. “Replacing Yellow Corn with Olive Cake Meal on growth Performance, plasma Lipid profile, and Muscle Fatty Acids Content in Broilers” forces on evaluating the effects of replacing corn with olive cake meal (OCM) in broiler's diet on performance. This manuscript may be resubmitted with revisions, details are as follows:

1.Line 24: p should be italic, and the similar errors should be checked and corrected.

2.Line 24: the full name the abbreviation that first appeared should be added.

3.Line 48-50: “the consumption of a certain amount of ……it is not just ineffective but also ineffective.” This sentence is ambiguous and needs to be reconsidered by the authors.

4.Line 53-59: the authors suggested that OCM is also high in lignin and contains xyloglucan, which limit its use in poultry feed. Whether OCM has passed special processing in this research? Does the author address the disadvantage of OCM?

5.Line 82-87: The sentence “According to reports, Saleh et al. …. resulting in the rise in poultry products” should be checked and corrected.

6.The authors should examine the manuscript carefully, for example Line 25: 35 days and Line 105: 0–10 d.

7.The authors need to explain why corn was chosen to be replaced by olive cake meal? What is the production of OCM?

  1. The authors suggest that 10% corn replacing with olive cake meal (OCM) is suitable for broiler diets, improved performance, reduced abdominal fat, plasma lipid profile, and FA content, why didn't the author choose 5% OCM?
  2. The discussion is not a statement of previous studies and a list of results. The author should focus on why corn should be replaced? what are the advantages and benefits of replaced corn with OCM? is it feasible to replace corn with OCM?why does replacing corn with 20% OCM result in lower performance for broilers? what is the purpose and significance of this research for production?

10.What is the purpose and significance of this research for measuring Vitamin E and MDA levels?

Author Response

Review report 1

The manuscript by Ahmed et al. “Replacing Yellow Corn with Olive Cake Meal on growth Performance, plasma Lipid profile, and Muscle Fatty Acids Content in Broilers” forces on evaluating the effects of replacing corn with olive cake meal (OCM) in broiler's diet on performance. This manuscript may be resubmitted with revisions,

Response: Thank you so much for your comments and we belief that your comments and advices will increase the scientific value of our manuscript.

details are as follows:

1.Line 24: should be italic, and the similar errors should be checked and corrected.

Response:         Thank you for your comment we corrected it L24.

2.Line 24: the full name the abbreviation that first appeared should be added.

Response:         Thank you for your comment we corrected as following (body weight (BW), feed intake (FI), and feed conversion ratio (FCR ) . L24.

3.Line 48-50: “the consumption of a certain amount of ……it is not just ineffective but also ineffective.” This sentence is ambiguous and needs to be reconsidered by the authors.

Response:         Thank you for your comment we corrected as following (The levels and balances of any new ingredient in a diet is another essential factor, which affect the feed consumption and feed conversion ratio) L48-49.

4.Line 53-59: the authors suggested that OCM is also high in lignin and contains xyloglucan, which limit its use in poultry feed. Whether OCM has passed special processing in this research? Does the author address the disadvantage of OCM?

Response:        Thank you for your comment we corrected as following (OCM is also high in lignin. It contains xyloglucan (a form of non-starch polysaccharide) linked to anti-nutritional effects in monogastric organs like poultry, limiting its use in poultry feed [6]. Moreover, the presence of significant amounts of anti-nutritional phenolics compounds such as tannins in the diet, when combined with protein and carbohydrates, may limit nutrient availability by reducing the performance of the digestive enzyme [7, 8]. Furthermore, one of the drawbacks of OCMs is their high humidity and fat content, making consumption and storage difficult [9]. However, some processing method of the olive oil extraction may improve the digestibility of OCM and reduction these anti-nutritional factors [2]. L53-59.

5.Line 82-87: The sentence “According to reports, Saleh et al. …. resulting in the rise in poultry products” should be checked and corrected.

Response:      Thank you for your comment we corrected as following (According to reports, Saleh et al. and Cayan & Erener [2,22], adding OCM by-products to poultry diets improved total serum protein, albumin, HDL cholesterol concentration, and an increased HDL/LDL ratio correlated with lower total cholesterol and LDL content. Furthermore, feeding broilers a dietary containing OCM increased the polyunsaturated fatty acids in the breast muscle [23]. L84-88.

6.The authors should examine the manuscript carefully, for example Line 25: 35 days and Line 105: 0–10 d.

Response:         Thank you for your comment we checked them, at L25 these are results after 35 days (the whole experiment period). At L 105 these the diets used , we used starter diets from 0–10 days; then fed on grower diets from 11–24 days and finally we fed on finisher diets from 25–35d).

7.The authors need to explain why corn was chosen to be replaced by olive cake meal? What is the production of OCM?

Response:         Thank you for your comment, we used OCM as sources of energy and chosen yellow corn because the level of protein and energy found in the OCM may be the same of corn as the following (Olive cake meal analysis (Crude protein; 8.2%, ME; 3330 Kcal/kg, Ca; 0.021%, A.vp. Phosphorus; 0.29%, Ether extract; 17.8%, Fiber; 11.55%). L 121-122.

  1. The authors suggest that 10% corn replacing with olive cake meal (OCM) is suitable for broiler diets, improved performance, reduced abdominal fat, plasma lipid profile, and FA content, why didn't the author choose 5% OCM?

Response:        Thank you for your comment replacing 5% and 10% may have the same results in different parameters but 10% will be better economic efficiency, in addition in our country corn in very expensive so replacing high level will be better than lower one.

  1. The discussion is not a statement of previous studies and a list of results. The author should focus on why corn should be replaced? what are the advantages and benefits of replaced corn with OCM? is it feasible to replace corn with OCM?why does replacing corn with 20% OCM result in lower performance for broilers? what is the purpose and significance of this research for production?

Response:           Thank you for your comment we added the following (Recently, alternative use of yellow corn for ethanol production consumes a significant amount of corn resulting in a large impact on corn price. This has had a major impact on poultry production. For example, in broiler production, yellow corn contents for ~60–70% of the diets. A decrease in the availability of corn and higher in the price for diets have a major effect on the broiler production. So we must found new alternative ingredients must be able to substitute for yellow corn partially or totally and must not have a negative effect on the growth performance of broilers. The utilization of olive by-products for example olive cake meal as broilers diets is a good way of recycling these waste products. But there is a need to formulate optimized rations for different broiler uses to avoid metabolic disorders caused by the unbalanced rations of energy and protein and to reduce the tasty factors which might limit feed intake and then the broilers growth performance that leads to low profitability. Most of the studies conducted done had focused on feeding OMC for broilers and its effect on growth performance and meat quality. During the current study we tried to found the best replaced levels of yellow corn by OCM which will not led any negative effects in the broiler production. L193-208.

10.What is the purpose and significance of this research for measuring Vitamin E and MDA levels?

Response:        Thank you for your comment, replaced OCM will increased unsaturated fatty acids and these fatty acids can be damage by oxidation, so if vitamin E increased this will protect the fatty acids oxidation and we can measuring this by decreasing MDA level

Reviewer 2 Report

Recommend the following modifications:

L11 - Make the "f" in "factories" lower case

L12 - The comma should go before the word "especially" and add the word "regarding" before "metabolizable energy".

In both the simple summary and abstract, the age and the start of the study breed/strain/sex of broiler should be included.

L23 - After "OCM in diet" add ", respectively"

L24 - Acronyms should be defined at first use unless these are allowed without definition by the journal.

L24-26 - Recommend removing these two sentences since they discuss non-significant data. L24 is particularly difficult to understand because it states that there "did not seem" to be differences, but P<0.05? The 2nd sentence is also not specific about the data it is referring to. It looks like these two sentences are verbatim from the results section, so be sure they are corrected or removed there also.

L27-28 - In both of these sentences, you discuss a trend, but state P>0.05 (implying non-significance). If not significant, take out these sentences, or correct P value statement if they are significant.

L29 - State the type of muscle being discussed in the sentence.

L32 - Define which "high level" you are referring to.

L20 and 34 - The word "performance" may be better describe as "growth" or a related term for more specific definition.

L39 - Change "replacing" to "using" or revise sentence. As it is structured, the sentence implies that you are replacing the low cost ingredients with something else. There are several cases in the abstract that read this way also, suggest an additional read through to fix any of these minor grammatical errors.

L44 - Define "OCM" at this first use in the main body of the manuscript. Needs definition separate from the abstract.

L46 - Change "include" to "are included in"

L48-50 - Please revise this sentence for content and grammar, I cannot interpret the point it is trying to convey and there are several repeated words.

L51 - Define "animals" and "aquaculture" species you are referring to. 

L52 - Suggest sticking with "OCM" instead of "olive pastry" for consistency. 

L59 - Do you mean high "moisture" content (rather than "humidity")?

L61 - Need to define "NSP" and why is it included next to protein and fat content?

L61-62 - Entire sentence needs revision, I cannot interpret the main points that are being conveyed clearly as written.

L62 - Change "As well as," to "Additionally,"

L71 and 77 - Must define "FA" and "GIT" at first use at these two places, respectively.

L71 - What animals were reported in the study you are referencing? 

L72 - Change "used" to "use"

L80 - What did these researchers compare to as their control?

L81 - Elaborate on what this author found about "efficiency in growth and meat quality", it is very general as written.

L82 - Change "Erener. [2,22]. Adding" to "Erener [2,22], adding"

L87 - Define "UFA" here since it is the first use of this term.

L104 - Remove "(" before "[NRC]"

L107-108 - Remove semicolon, add ", respectively", and replace with period. Capitalize "The" in the new sentence.

L108 - Change "proved from" to "provided by"

L109 - ad libitum should be italic

Table 1 is not referred to in the text and must be included in an adjacent paragraph to refer to the diet composition.

L156 - Change "lustrated" to "illustrated" or "shown"

L158-160 - L158 is difficult to understand because it states that there "did not seem" to be differences, but P<0.05? Please clarify. The 2nd sentence is also not specific about the data it is referring to. It looks like these two sentences are verbatim from the abstract, so be sure they are corrected there also.

L175 - I believe you mean P<0.05 since you state that the findings are significant. Please correct. 

L176-177 - I believe you mean P<0.05 since you state that the findings are significant. Please correct. Also, please specify which OCM group was the highest, that is not stated currently. 

Figure 1 - This figure could be taken out and and the date included in two additional rows of table 4. Please do this to make the data presentation more concise or provide justification for while this data is singled out as separate figures and not in a table.

L210 - Change "did not affect" to "were not affected"

L222-223 - Change to "p<0.05" since I believe this was the intention because you are talking about significant data.

L246-247 - Change "fee" to "feed" and remove statements about probiotics since they are not the focus of the current study and statements seem out of place at the very end of the manuscript.

L250-251 - Change "replacing" to "replaced" and "performance" to "growth performance". Provide more specific details in conclusions on the impact of OCM on "plasma lipid profile, and FA content" since it is not clear as written.

Author Response

Review report 2

L11 - Make the "f" in "factories" lower case

Response:         Thank you for your comment, we corrected it L11

L12 - The comma should go before the word "especially" and add the word "regarding" before "metabolizable energy".

Response:         Thank you for your comment, we corrected it L12

In both  the simple summary and abstract, the age and the start of the study breed/strain/sex of broiler should be included.

Response:         Thank you for your comment, we added it L 16, L22.

L23 - After "OCM in diet" add ", respectively"

Response:         Thank you for your comment, we added it L24

L24 - Acronyms should be defined at first use unless these are allowed without definition by the journal.

Response:         Thank you for your comment, we did it L26

L24-26 - Recommend removing these two sentences since they discuss non-significant data. L24 is particularly difficult to understand because it states that there "did not seem" to be differences, but P<0.05? The 2nd sentence is also not specific about the data it is referring to. It looks like these two sentences are verbatim from the results section, so be sure they are corrected or removed there also.

Response:         Thank you for your comment, we corrected it and removed the difficult understand sentences

L27-28 - In both of these sentences, you discuss a trend, but state P>0.05 (implying non-significance). If not significant, take out these sentences, or correct P value statement if they are significant.

Response:         Thank you for your comment, we corrected it L27-28.

L29 - State the type of muscle being discussed in the sentence.

Response:         Thank you for your comment, we added it as following (The breast muscle (Musculus pectoralis superficialis) L29.

L32 - Define which "high level" you are referring to.

Response:         Thank you for your comment, we added it L33.

L20 and 34 - The word "performance" may be better describe as "growth" or a related term for more specific definition.

Response:         Thank you for your comment, we corrected it L 21, L 35.

L39 - Change "replacing" to "using" or revise sentence. As it is structured, the sentence implies that you are replacing the low cost ingredients with something else. There are several cases in the abstract that read this way also, suggest an additional read through to fix any of these minor grammatical errors.

Response:         Thank you for your comment, we corrected it L40.

L44 - Define "OCM" at this first use in the main body of the manuscript. Needs definition separate from the abstract.

Response:         Thank you for your comment, we added it L 45.

L46 - Change "include" to "are included in"

Response:         Thank you for your comment, we added it L47.

L48-50 - Please revise this sentence for content and grammar, I cannot interpret the point it is trying to convey and there are several repeated words.

Response:         Thank you for your comment, we revised it L 48-49.

L51 - Define "animals" and "aquaculture" species you are referring to. 

Response:             Thank you for your comment, we added it as following (It also helps reduce ration costs as well as the animals (sheep), poultry, and aquaculture (Cyprinus carpio) feeding cost. The variability of its chemical composition is one of the most critical limitations of olive pastry [5]. L50-52.

L52 - Suggest sticking with "OCM" instead of "olive pastry" for consistency. 

Response:         Thank you for your comment, we corrected it L52.

L59 - Do you mean high "moisture" content (rather than "humidity")?

Response:         Thank you for your comment, we corrected it L59.

L61 - Need to define "NSP" and why is it included next to protein and fat content?

Response:         Thank you for your comment; we corrected it as following (Additionally, the OCM, which, as well as a high content of Non starch polysaccharides (NSP), has a high nutritional value (Fat, 13-15%, proteins, 9-10%) [10-12]. ) L62-63.

L61-62 - Entire sentence needs revision, I cannot interpret the main points that are being conveyed clearly as written.

Response:         Thank you for your comment, we corrected it.

L62 - Change "As well as," to "Additionally,"

Response:         Thank you for your comment, we corrected it L 64.

L71 and 77 - Must define "FA" and "GIT" at first use at these two places, respectively.

Response:         Thank you for your comment, we corrected it L72, 78.

L71 - What animals were reported in the study you are referencing? 

Response:         Thank you for your comment, we added it as following (broilers tissue fatty acids ) L 72.

L72 - Change "used" to "use"

Response:         Thank you for your comment, we corrected it L74.

L80 - What did these researchers compare to as their control?

Response:         Thank you for your comment, we added it as following (Papadomichelakis et al. recently demonstrated that broilers fed 50-80g/kg of dried olive pulp had achieved more efficiency in growth and meat quality compared the control group [21].) L82-84.

L81 - Elaborate on what this author found about "efficiency in growth and meat quality", it is very general as written.

Response:         Thank you for your comment, we observed it as following (had better growth performance (body weight and feed conversion ratio) and meat quality (reduced oxidative stability and better meat color)

L82 - Change "Erener. [2,22]. Adding" to "Erener [2,22], adding"

Response:         Thank you for your comment, we corrected it L85.

L87 - Define "UFA" here since it is the first use of this term.

Response:         Thank you for your comment, we corrected it L88.

L104 - Remove "(" before "[NRC]"

Response:         Thank you for your comment, we corrected it L107.

L107-108 - Remove semicolon, add ", respectively", and replace with period. Capitalize "The" in the new sentence.

Response:         Thank you for your comment, we corrected it as following (for male broilers, with a three-phase feeding system (starter diets from 0–10 days, grower diets from 11–24 d and finisher diets from 25–35d respectively).

L108 - Change "proved from" to "provided by"

Response:         Thank you for your comment, we corrected it L111.

L109 - ad libitum should be italic

Response:         Thank you for your comment, we corrected it L112.

Table 1 is not referred to in the text and must be included in an adjacent paragraph to refer to the diet composition.

Response:        Thank you for your comment, we corrected it L109.

L156 - Change "lustrated" to "illustrated" or "shown"

Response:         Thank you for your comment, we corrected it L159.

L158-160 - L158 is difficult to understand because it states that there "did not seem" to be differences, but P<0.05? Please clarify. The 2nd sentence is also not specific about the data it is referring to. It looks like these two sentences are verbatim from the abstract, so be sure they are corrected there also.

Response:         Thank you for your comment, we corrected it as following (Broilers fed dietary replaced corn with 5% or 10% OCM had better body weight than 20% OCM replaced group (p < 0.040). Additionally, replaced 5 or 10% have better feed conversion ratio than control or 20% groups.  ) L 159-161.

L175 - I believe you mean P<0.05 since you state that the findings are significant. Please correct. 

Response:         Thank you for your comment, we corrected it.

L176-177 - I believe you mean P<0.05 since you state that the findings are significant. Please correct. Also, please specify which OCM group was the highest, that is not stated currently. 

Response:         Thank you for your comment, we corrected it as following (Except for total cholesterol in the plasma, this decreased significantly (p > 0.038) by replacing corn with 5, 10 and 20% OCM in broilers. In addition, the highest HDL value (p > 0.022) was found in 20% OCM replaced bird group.) L178-180.

Figure 1 - This figure could be taken out and and the date included in two additional rows of table 4. Please do this to make the data presentation more concise or provide justification for while this data is singled out as separate figures and not in a table.

Response:         Thank you for your comment, remove figure and added the data in the table.4.

L210 - Change "did not affect" to "were not affected"

Response:         Thank you for your comment, we corrected it L206.

L222-223 - Change to "p<0.05" since I believe this was the intention because you are talking about significant data.

Response:         Thank you for your comment, we corrected it L218.

L246-247 - Change "fee" to "feed" and remove statements about probiotics since they are not the focus of the current study and statements seem out of place at the very end of the manuscript.

Response:         Thank you for your comment, we corrected it L242- 244.

L250-251 - Change "replacing" to "replaced" and "performance" to "growth performance". Provide more specific details in conclusions on the impact of OCM on "plasma lipid profile, and FA content" since it is not clear as written

Response:         Thank you for your comment, we revised it L 246-248.

Round 2

Reviewer 1 Report

I think this manuscript meets the requirement of publish

Author Response

Author's Reply to the Review Report (Reviewer 1)

Comments and Suggestions for Authors

I think this manuscript meets the requirement of publish

Response: Thank you for your comments and we belief that your comments will increase the scientific value of our manuscript.